# An improved nuclei isolation protocol from leaf tissue for single-cell transcriptomics

**Gabriela Madrid[1,2], Gabriel Angelo Saraiva Raimundo[2], Fabian Andres Reyes[2], Edgard Augusto de Toledo Picoli[2,3], Marcio F. R. Resende Jr.[1,2]\*, Kelly M. Balmant[2]\***

**1** Plant Molecular and Cellular Biology Program, University of Florida, Gainesville, Florida, United States of America, **2** Horticultural Sciences Department, University of Florida, Gainesville, Florida, United States of America, **3** Departamento de Biologia Vegetal, Universidade Federal de Viçosa, Viçosa, State of Minas Gerais, Brazil

\* balmant@ufl.edu (KMB); mresende@ufl.edu (MFRR)

## Abstract

The study of plant biology has traditionally focused on investigations conducted at the tissue, organ, or whole plant level. However, single-cell transcriptomics has recently emerged as an important tool for plant biology, enabling researchers to uncover the expression profiles of individual cell types within a tissue. The application of this tool has revealed new insights into cell-to-cell gene expression heterogeneity and has opened new avenues for research in plant biology. A critical step in the successful application of single-cell and single-nuclei RNA-seq (scRNA-seq and snRNA-seq) is the isolation of individual cells or nuclei, respectively, from tissue to recover their transcriptional profile. A critical step during nuclei isolation for snRNA-seq studies is Fluorescent-Activated Cell Sorting (FACS). During this step, nuclei stained with DAPI (4′,6-diamidino-2-phenylindole) can be sorted and separated from cell debris and organelles. Leaf tissue presents a unique challenge due to its high content of chloroplasts, which can interfere with obtaining high-quality results. Because DAPI can also bind to the plastid genome, these organelles will be sorted as nuclei. Thus, in tissues with a high content of chloroplasts, we have a high contamination of these organelles and an overestimation of the number of nuclei. In this study, we introduce a straightforward alternative method for isolating nuclei from *Zea mays* leaves with reduced chloroplast contamination. By effectively removing chloroplasts during the FACS step of our protocol, using the autofluorescence from the chloroplasts, we achieved improved alignment of reads to the genome and transcriptome. Our enhanced protocol offers a valuable solution for applying snRNA-seq in tissues with a high content of chloroplasts.

## Introduction

Single-cell RNA sequencing (scRNA-seq) has emerged as a valuable tool in plant biology, enabling a detailed exploration of gene expression at the cellular level. From

**Data availability statement:** All raw and processed sequencing data generated in this study have been submitted to the NCBI Gene Expression Omnibus (GEO; https://www.ncbi.nlm.nih.gov/geo/) under accession number GSE297213. Custom scripts are available at https://github.com/Resende-Lab.

**Funding:** DOE Grant No. DE-SC0023082 (to K.M.B.); USDA NIFA SCRI Grant No. 2018-51181-28419 and 2022-51181-38333 (to M.F.R.R).

**Competing interests:** The authors have declared that no competing interests exist.

unraveling intricate signaling pathways and developmental trajectories to elucidating the molecular dynamics of stress responses, scRNA-seq offers several research opportunities [1–5]. Furthermore, this technology has the power to identify previously undiscovered cell types [6], decipher cell type-specific gene regulatory networks [7], and gain a deeper understanding of the mechanisms driving cell differentiation [8]. In the years to come, as scRNA-seq is further integrated into plant biology research, we anticipate it will enhance our comprehension of cellular diversity, plant development, and environmental responses.

The success of scRNA-seq is dependent on the acquisition of high-quality cells. The quality of data obtained during scRNA-seq procedures significantly impacts the reliability and robustness of subsequent analyses. Attention must be directed toward several critical aspects to extract meaningful insights from these datasets. First, the choice of tissue processing and cell isolation methods is crucial. The quality of the biological material and the preservation of the cells significantly influence the outcomes of scRNA-seq. The handling of plant tissues, from dissociation to library preparation, must minimize bias and maintain the integrity of cellular RNA [9]. Moreover, the selection of appropriate equipment and protocols for the sequencing process itself is vital. Factors such as read depth, library complexity, and sequencing quality directly affect the information extracted from the dataset [10].

A dataset containing suboptimal quality data, represented by the high presence of "low quality" cells, can impact the total number of cells obtained. This can yield inaccurate biological interpretations, potentially leading to the mischaracterization of cell types, gene expression profiles, and cellular functions [11]. This is exacerbated by a reduction in statistical power [12], hindering the detection of subtle gene expression variations or the identification of rare cell types.

In plants, an approach aiming to facilitate tissue dissociation and reduce the stress response is using nuclei over protoplasts for the cell isolation step to perform single-nuclei RNA-seq (snRNA-seq). The nuclei isolation protocols currently in use [9,13] are adapted versions of the protocols historically used for estimating genome size, cell cycle analysis, and ploidy level by flow cytometry [14]. For single-cell transcriptomics, the protocols have been simplified to reduce the use of reagents that can interfere with the library preparation step and to reduce the time to process samples. Briefly, the tissue of interest is first homogenized in the isolation buffer. After, this crude extract is filtered to remove intact cells and large debris and then centrifuged at low speeds to pellet the nuclei and further remove cell debris and contaminants, followed by a step to enrich nuclei with Fluorescent Activated Cell Sorting (FACS). A major challenge of nuclei isolation is to minimize the leaking of nuclear content due to damage to the weakened nuclear membrane. This leads to loss of nuclear integrity and, more importantly, loss and leakage of transcripts [13]. Another challenge of nuclei purification is the complete removal of the cytoplasm and correct separation from other organelles without damaging the nuclear membrane [15].

Both nuclei leakage and incomplete removal of cytoplasm and organelles can have negative effects as they contribute to the presence of ambient RNA, which is extraneous RNA molecules present in the cellular environment, distinct from the

mRNA content of the individual cell being profiled and can introduce unwanted noise and technical variability into snRNA-seq data [16]. For droplet-based methods, where the individual cells or nuclei are captured in microfluidic droplets containing the reagents for library preparation [17], it is assumed that each droplet contains only RNA from an intact single cell or nucleus. The presence of ambient RNA deviates from this assumption as it can be present in both empty droplets and in droplets containing a cell. These cell-free RNAs will be mistakenly included as part of an intact cell's RNA, confounding the identity of the original cell. This may lead to erroneous merging of two distinct cell populations during downstream analyses [11].

Leaf tissues present a different level of difficulty in purifying intact nuclei due to the high presence of chloroplasts in their cells. Leaf cells can contain hundreds of chloroplasts with copies of the plastid genome (cpDNA) in the 10,000 range [18]. Chloroplasts have been shown to be hard to purify away from nuclei, and attempts to remove them by using stringent detergents usually result in damaging the nuclear membrane [15,19]. Notably, contamination with cpDNA during DNA isolation for genome sequencing has been shown to diminish sequencing coverage of the nuclear genome, with a portion of lost reads aligning to the cpDNA [18,20,21]. Recent findings emphasize that, despite the chloroplast genome's limited gene content, transcripts originating from chloroplasts constitute most of the mRNA pool in leaf cells [22], accentuating a distinct chloroplast-nuclear asymmetry in RNA expression. This asymmetry introduces potential challenges for RNA-seq outcomes. The entire plastome is transcribed in photosynthetic tissue, and the plastid transcriptome can be effectively isolated from total transcriptomes obtained from RNA-seq with considerable read depth [23]. This is possible because the precursor transcripts from the chloroplasts are also polyadenylated [24], similar to nuclear mRNA, although for plastid transcripts, this mechanism is part of the RNA degradation pathway [25]. This would allow for the amplification of plastid transcripts along with nuclear genes during library preparation.

The presence of chloroplast transcripts poses a concern during library preparation for snRNA-seq, diminishing sequencing coverage and contributing to the pool of cell barcodes with "low quality" data. To address this issue, we present an enhanced protocol for nuclei isolation utilizing a FACS strategy for chloroplast removal prior to library preparation. With this approach, we could minimize chloroplast contamination, obtaining intact and pure nuclei from maize leaves without the need for stringent detergents that damage the nuclei, a common approach. In addition, our proposed method does not increase costs or processing time, which are both critical factors for selecting the sample preparation process for snRNA-seq. Using this protocol, we enhanced the genome and transcriptome alignment rates to levels concordant with high-quality snRNA-seq libraries and detected a higher number of genes.

Moreover, we improved the overall quality of the cells that passed through our first quality control step. Our approach generated no bias in cell clustering and cell identification. These results suggest that our protocol has the potential to enhance single-nuclei RNA sequencing experiments in leaves.

## Materials and methods

### Sample collection

The inbred B73 maize (*Zea mays L.*) was used for this study. Seeds were planted in potting mix (peat moss, perlite, vermiculite mix) and grown in a greenhouse at 24 °C and ambient humidity under natural light at the University of Florida, Gainesville, Florida, in the Fall of 2021. Plants were grown until the V5 stage, when samples were collected from the 4th fully extended leaf and immediately used for nuclei isolation. The midsection of the leaf was dissected and separated into sections of about 1 cm x 1 cm. One leaf from three plants was pooled for nuclei isolation.

### Histological sections

Leaf sections from V5 leaves from maize plants were collected and fixed in FAA50 (formalin, acetic acid, ethanol 50%; 5: 5: 90 v/v) for 24 h and subsequently rinsed and stored in ethyl alcohol 50%. The samples were dehydrated in ethylic series

and included in glycol methacrylate (Technovit 7100, Kulzer). The leaves were sectioned transversely (8 µm thick) in a manual rotary microtome (RM 2235, Leica). The sections were stained with Toluidine Blue pH 4.0 for 1 min and mounted with synthetic resin (Permount, Fisher). All the histological analysis and photographic documentation were performed with an Olympus BX51 microscope equipped with a U-Photo system.

## Nuclei suspension preparation

To obtain nuclei from the leaf tissue collected, we used a previously successful protocol by Conde et al. (2021) [13]. Briefly, the tissue was homogenized with a razor blade using the Nuclei Isolation Buffer to release the nuclei. Then, the nuclei were separated from cellular debris by sequential filtering and centrifugation steps. The nuclei were then stained with 4',6-diamidino-2-phenylindole (DAPI). All steps were performed on ice inside a cold room.

## Nuclei purification by fluorescent-activated cell sorting

After isolation, we used a FACS strategy for clean-up and enrichment of the nuclei using a BD FACSAriaTM IIU/III upgraded cell sorter at the Interdisciplinary Center for Biotechnology Research at the University of Florida (Gainesville, FL, USA. RRID:SCR_019119). We used a double-filter strategy to negatively select autofluorescent chloroplast and positively select DAPI-stained nuclei (Fig 1). The Peridinin-Chlorophyll-Protein (PerCP) filter, excited by the 488 nm blue laser and captured with a 670/30 nm bandpass filter, was used to remove the events that emitted in this range. After this, the DAPI-positive population was selected according to emission in the 450/50 nm bandpass filter and then based on size and granularity using the forward versus side scatter (FSC vs SSC) plot. Finally, the selected population corresponding to nuclei was sorted, aiming for 40,000 nuclei for library preparation. We also used a single filter strategy as described previously [8] to only positively select DAPI-stained nuclei. This sample was used as a control to test the efficacy of our method. The presence of chloroplasts and nuclei quality was assessed using a Zeiss LSM 800 confocal microscope at the Interdisciplinary Center for Biotechnology Research at the University of Florida (Gainesville, FL, USA. RRID:SCR_019146).

## snRNA-seq library preparation and sequencing

We followed the Single Cell v3.1 Dual Index Gene Expression protocol from 10x Genomics. A total of 20,000 sorted nuclei were used for cDNA synthesis and sample amplification following the manufacturer's instructions. Sequencing was performed at the Interdisciplinary Center for Biotechnology Research at the University of Florida (Gainesville, FL, USA. RRID:SCR_019145) using an Illumina NovaSeq6000. The sequencing was performed using a standard program as follows: 28 bp (Cell barcode and UMI) for read 1, 90 bp (cDNA) for read 2, 10 bp for I7 Index, and 10 bp for I5 Index. We sequenced a total of 288,909,642 reads for the library prepared with the FACS double-filter strategy and 154,036,205 reads for the library prepared with the FACS single-filter strategy. The sequencing quality control metrics for each library were the following: Library prepared with double-filter strategy (Sequencing saturation: 87%; Q30 Bases in barcode: 93.4%; Q30 Bases in RNA Read: 89%; Q30 in UMI: 91.6%) and Library prepared with single-filter strategy (Sequencing saturation: 86.1%; Q30 Bases in barcode: 92.9%; Q30 Bases in RNA Read: 84.8%; Q30 in UMI: 91.4%).

## Computational workflow for scRNA-seq data analysis

The reference *Zea mays* genome (B73 RefGen_v5) does not include the sequences for mitochondria and chloroplast-encoded genes. For our analysis, we downloaded the mitochondria and chloroplast reference genomes and annotation files from Ensembl (ftp://ftp.ensemblgenomes.org/pub/plants/release-42) and concatenated both to the B73v5 genome. We used the CellRanger pipeline (v7, 10X Genomics) to first build a reference using the "cellranger mkref" and the "cellranger count" function to generate a raw count matrix. This gene expression matrix was used for filtering, cell clustering,

## 1) Sample collection

Leaves samples were collected from V5 leaves from maize plants.

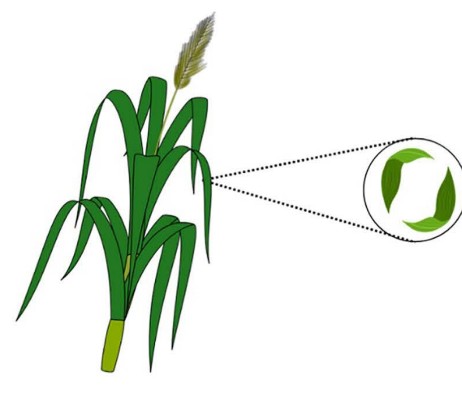

## 2) Nuclei suspension preparation

Nuclei suspension was obtained using the protocol by Conde et al. (2021) [8]. Nuclei were stained with DAPI.

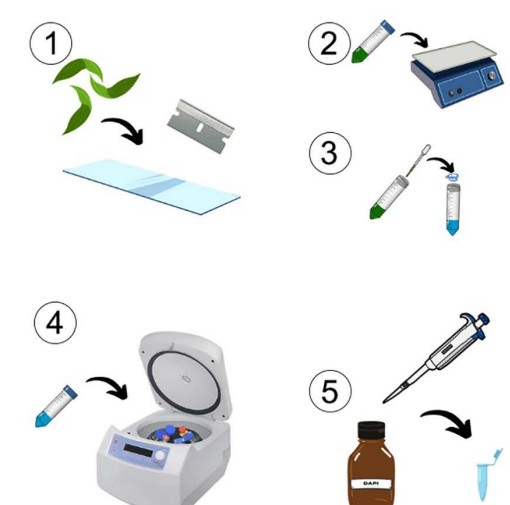

## 3) Nuclei purification by fluorescent-activated cell sorting

Nuclei suspension was cleaned-up and enriched by FACS strategy using double filter to positively select DAPI-stained nuclei and negatively select autofluorescent chloroplast.

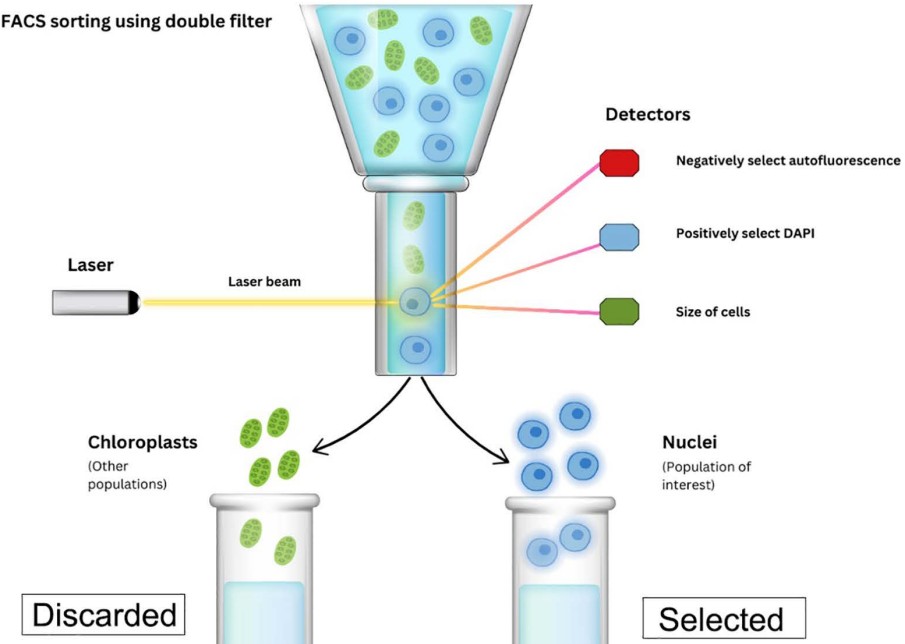

**Fig 1. Nuclei isolation workflow using the FACS double-filter strategy.** Workflow diagram showing the nuclei isolation from maize leaves for snRNA-seq library.

and cell type identification analysis using the Seurat R package (v4.1.1) pipeline [26]. We normalized the total number of reads of the two libraries. We randomly selected the same number of reads in the current library as in the one being compared using the seqtk processing tool. With these normalized reads, we built the count matrix as mentioned above.

Only nuclei with the number of UMI ≥ 500, the number of genes per nucleus ≥ 250, < 5,000, and a plastid percentage ≤ of 0.5% were considered for downstream analysis. Integration and clustering of the two snRNA-seq datasets were carried out at a resolution of 0.5. Uniform Manifold Approximation and Projection (UMAP) was used to reduce the complexity and visualize the data in two dimensions. Thirty components were consistently selected for dimensionality reduction across all samples.

Cell-type identification was performed by examining the expression pattern of known cell-type-specific markers from the literature (S1 Table and S2 Fig).

### Generation of quality metrics

In addition to the standard quality assessment for scRNA-seq, which includes transcripts per nucleus and genes detected per nucleus, we determined the proportion of transcripts from chloroplast-encoded genes. This metric is used to identify if there is a large amount of chloroplast contamination. For this, we used the "PercentageFeatureSet" function from Seurat (v4.1.1) [26] with the "Zmcp" pattern that identifies genes corresponding to the chloroplast. This function calculates the column sum of the matrix present in the count's slot for features containing the provided pattern, divided by the column sum for all features times 100. Cells with a chloroplast percentage higher than 0.5% were removed for further analysis.

### Nuclei and chloroplasts counting

Mid-sections of the V5 stage of B73 leaves were used to isolate nuclei as described above. Half of the sample was sorted using the double filter strategy (positively select DAPI-stained nuclei and negatively select chloroplasts by autofluorescence), and half of the sample was sorted using a single filter (only positively select DAPI-stained nuclei) as a control. After sorting, 10 µL of the sorted events from each sample were loaded into a DHC-N01-2 Neubauer Improved. Images were taken with a microscope Olympus BX51 microscope with a 10X objective. The first, second, third, fourth, and central quadrants were photographed with a bright field and filters for DAPI and GFP. The exposure times for all image filters were 0.536ms, 650ms, and 650ms, respectively, for bright field, DAPI, and GFP, respectively. The GFP filter was deployed as a way to count chloroplasts since its autofluorescence emits red light for these conditions. The average of events corresponding to nuclei and chloroplasts of each quadrant was used to calculate the number of events per milliliter using the following formula:

$$\frac{Events}{mL} = Average\ event\ counts\ per\ sqaure\ x\ dilution\ factor\ x\ 10^4$$

(1)

### Quantitative real-time PCR

Nuclei samples were obtained as described above and were frozen after sorting. Total RNA was extracted from the frozen aliquots with the RNeasy® minikit 74104. A total of 12 ng of total RNA was used for cDNA synthesis, and RT-qPCR was performed using the Luna® universal One-step RT-qPCR (E3005). We selected two target genes that are specific to the chloroplast genome (*psbA* and *atpA*) [27] and two nuclear genes (*EIF1α* and *Ubq7*) that were used as reference genes. Gene expression levels were assessed in three independent biological replicates. The relative expression of each sample was determined after normalization to *EIF1α* and *Ubq7* using the $2^{-\Delta\Delta Ct}$ method. Means of different groups were compared and analyzed using a Mann-Whitney test. Differences were reported as statistically significant when $P$-value < 0.05. Sequences of gene-specific primers used in the experiment are found in the S2 Table.

## Results

Protocols for nuclei isolation for single-cell transcriptomics must consider the intricacies of each tissue being studied to obtain high-quality nuclei and assure data quality. Here, we have developed an improved isolation protocol for leaf tissue and tested it in mature maize leaves. We incorporated a stringent FACS strategy that accounts for the high presence of plastids in these samples, which can affect data usability and introduce biases. Data collected during FACS allows for the extraction of pure and intact nuclei and smooth *in-silico* processing of snRNA-seq data.

The proposed FACS strategy leverages the autofluorescence from chlorophyll to detect the presence of plastids in the leaf samples. For this, the first step was to identify the particles that emit wavelengths in the red-fluorescence range using the PerCP-Cy5.5 filter. In a histogram plot of the number of events vs. PerCP-Cy5.5 relative fluorescence (Fig 2A), two overlapping populations can be observed with two distinctive peaks at around the $10^4$ value. The particles above the $10^4$ value correspond to events positive for chlorophyll fluorescence and were determined to be contaminating chloroplasts in the nuclei suspension. Out of 2.5 million initial events, approximately 1.4 million events were gated as a PerCP-negative population to continue with the FACS strategy (Fig 1A), and from this population of events, DAPI-stained particles were selected for further analysis (~ 15,000 events) (Fig 2B and 2C). The DAPI-positive events were gated for DNA content using a DAPI histogram. Nuclei from maize leaves are expected to show two peaks corresponding to 2n and 4n ploidy level nuclei (Fig 2D) [28]. Next, we estimate the size and granularity (particle's internal structure) of the nuclei by using forward (FSC) and side scatter (SSC) dot plots, respectively. The nuclei population will have relatively similar size and granularity; therefore, we gated the highest-density nuclei population for further analysis (Fig 2E and 2F). This step also serves to remove doublets and further debris contamination. After this step, the number of DAPI-positive events decreases to ~5,000 events (Fig 2G) and corresponds to the final events to be sorted (Fig 2H). At this point, we obtain nuclei that contain considerably less cellular debris and plastid contamination and are, therefore, of greater quality for use in single-cell RNA sequencing.

Microscope observation was carried out to inspect the nuclei suspension stained with DAPI before and after FACS (Fig 3). We observed both the presence of nuclei and chloroplasts in the suspension before sorting (Fig 3A–3C), compared with after sorting (Fig 3D–3F), with a considerable number of chloroplasts found. We can corroborate the elevated presence of chloroplasts in the dot plot of PerCP-Cy5.5 vs. DAPI fluorescence (S3 Fig, black dots). The events positive for chlorophyll fluorescence are found in the upper left quadrant. After sorting, the population of DAPI-positive nuclei is enriched (S3 Fig, orange dots), and the population of PerCP-positive events is reduced, indicating that our strategy yields clean and intact nuclei.

Our first aim for developing a protocol that decreases plastid contamination of nuclei suspensions for scRNA-seq was to improve the overall sequencing metrics. We had previously generated a snRNA-seq library using a previously published nuclei isolation protocol [13] that is based on a single filter strategy during the FACS step that only positively selects the DAPI-stained events. We found 54% alignment to the reference genome (Table 1), which is lower than expected of at least 70% genome alignment [29,30]. This is a key indicator of high-quality libraries, and a low value for alignment indicates potential problems with RNA quality. Upon further exploration, by aligning the unmapped reads to the maize chloroplast genome, we determined that an important source of unmapped reads corresponded to chloroplast-encoded genes. A snRNA-seq second library was generated using the double filter strategy to positively select DAPI-stained nuclei and negatively select autofluorescent chloroplasts.

By decreasing the chloroplast contamination using this double filter strategy, we found 68% of the reads aligned to the genome (Table 1). The reads confidently mapped to the genome increased to almost 64% from 50%, indicating a positive and substantial change (Table 1). The percentage of reads aligned to the transcriptome also showed an almost 13-percentage-point increase compared to the previous method (Table 1).

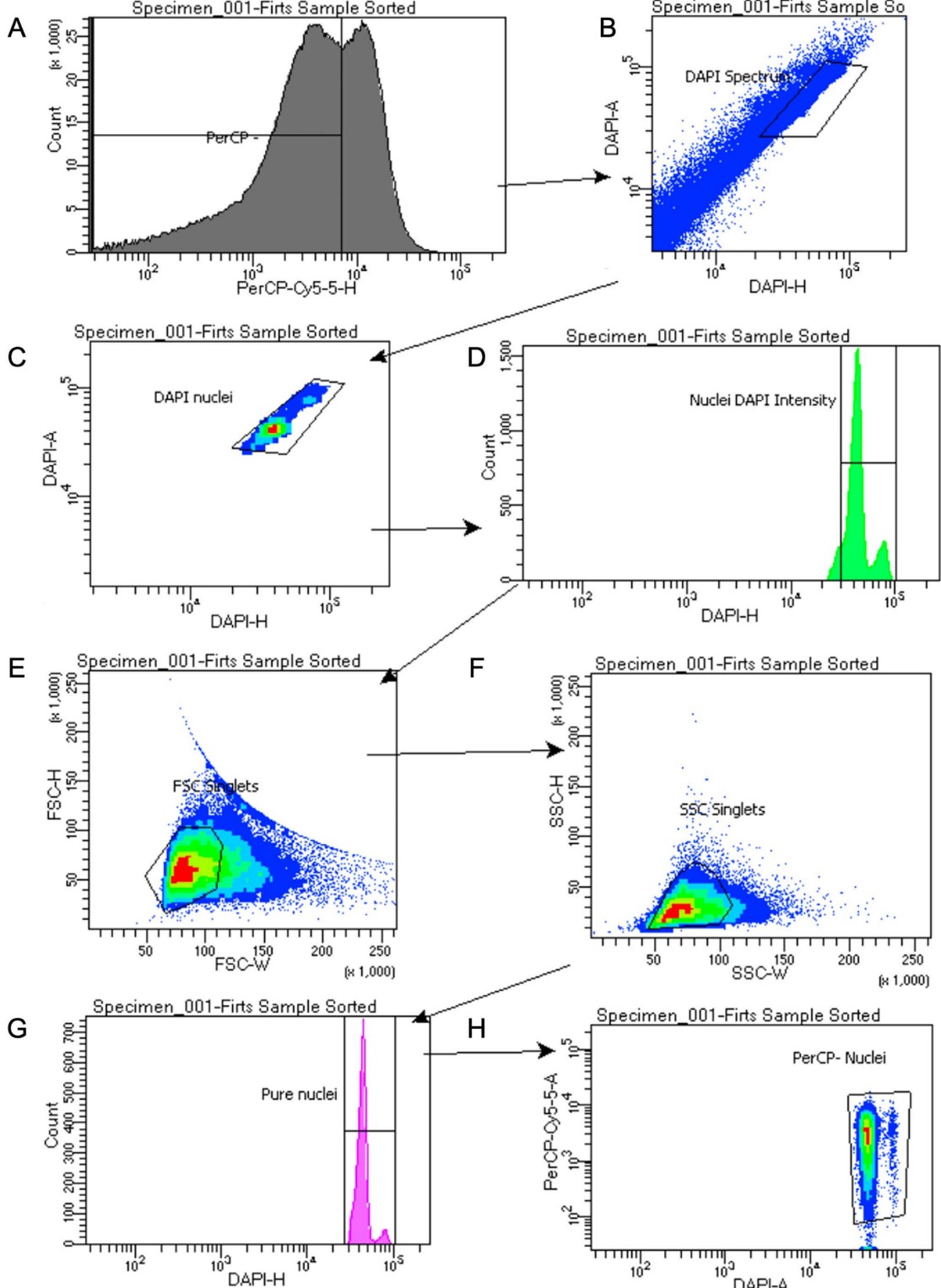

**Fig 2. Fluorescent Activated Nuclei Sorting processing using nuclei suspension from maize leaves.** (A) PerCP-negative events were selected by creating a gate in the PerCP histogram and selecting the particles below the $10^4$ value. (B)(C) DAPI area vs DAPI height was used to select DAPI-stained particles. (D) Histogram of the number of events positive for DAPI staining. (E)(F) Forward scatter and side scatter height vs area were used to

select for singlet events. (G) Histogram showing the number of positive events for DAPI after doublet removal. (H) PerCP area vs. DAPI area showing pure nuclei events used for sorting. Colors in the dot plot indicate population density, with red the highest and blue the lowest.

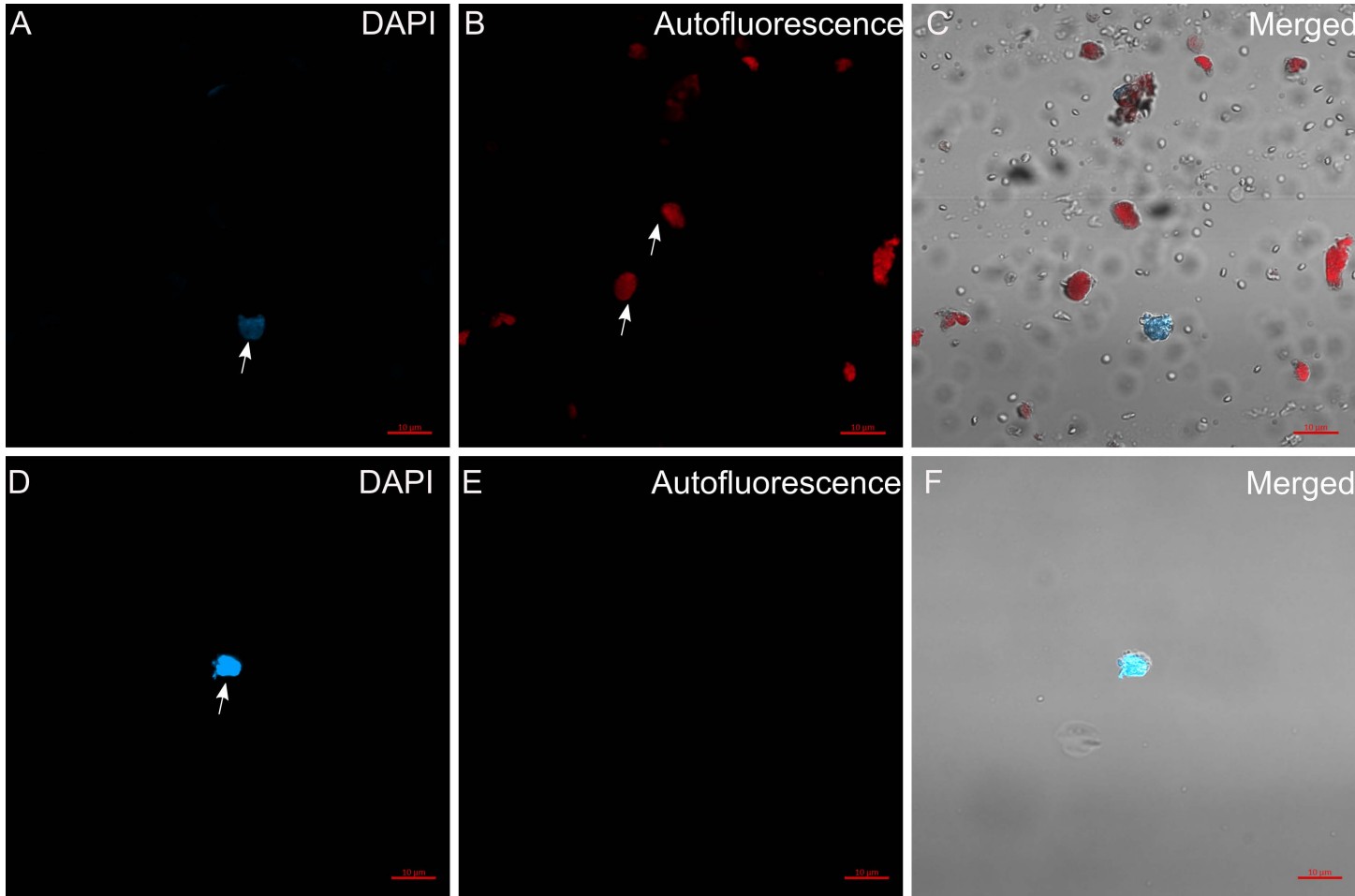

**Fig 3. Assessment of nuclei suspension.** Nuclei were examined under the microscope to determine nuclear integrity and the presence of chloroplasts. A-C) Pre-sorting confocal microscopy images of A) DAPI-stained nuclei, B) autofluorescent chloroplasts, and C) merged fluorescence and brightfield images. D-F) Post-sorting confocal microscopy images of D) DAPI-stained nuclei, E) Auto-fluorescent chloroplasts, and F) Merged fluorescence and brightfield images.

**Table 1. Summary of sequencing metrics for libraries generated with and without chloroplast removal.**

|  | FACS double filter strategy | FACS single filter strategy |
|---|---|---|
| Reads Mapped to Genome (%) | 68 | 54.1 |
| Reads Mapped Confidently to Genome (%) | 63.9 | 50.1 |
| Reads Mapped Confidently to Intergenic Regions (%) | 11.1 | 9.8 |
| Reads Mapped Confidently to Exonic Regions (%) | 52.7 | 40.3 |
| Reads Mapped Confidently to Transcriptome (%) | 49.9 | 37.4 |

A second aim was to improve library quality and specifically to reduce the number of low-quality potential nuclei in the library resulting from a high percentage of chloroplast-related transcripts. In general, using the improved strategy proposed here, we were able to improve the number of high-quality potential nuclei present in the library. The library generated with our improved protocol showed an increase in the mean number of Unique Molecular Identifiers (nUMIs) per nucleus of 201.97% (mean of library with filter = 5,678.44 nUMIs per nucleus, mean of library without filter = 1,880.44 nUMIs per nucleus) (Fig 4A). We also observed an increase in the mean of the number of genes (nGenes) per nucleus in our improved method by 125.05% (mean of library with filter = 2,611.65 nGenes per nucleus, mean of library without filter = 1,160.5 nGenes per nucleus) (Fig 4B). Furthermore, we observed an improvement in the number of genes detected. We detected 16,886 genes in the sample prepared with our improved method using the double-filter strategy and 12,607 genes in the sample prepared with the single-filter strategy (S1 Fig). In terms of the presence of potential nuclei with a high percentage of chloroplast-encoded transcripts, the use of our method reduced the number of cell barcodes above the quality threshold, which is a chloroplast-gene percentage higher than 0.5% (Fig 4C). After applying the threshold, 479 barcodes (which corresponds to 19% of the total number of barcodes) were removed from the dataset in the sample generated using the double filter strategy, whereas 844 barcodes (which corresponds to 33% of the total number of barcodes) were removed from the dataset of the sample prepared with the single filter strategy.

In addition to determining quality metrics, we also performed clustering analysis and cell type identification to determine if our strategy induced any bias in the cell types recovered compared to the previous study. To identify cell types, we used cell-type-specific markers (S1 Table and S2 Fig). We observed similar clustering results between both libraries, and we were able to confidently identify most of the known cell types for leaf tissue (Fig 5). Upon determining the percentage of cells corresponding to each cell type, we observed differences in the number of cells corresponding to the mesophyll and bundle sheath cells (Tables 2 and S3). The percentage of mesophyll cells in the snRNA-seq library generated with the single filter strategy was higher (58%) compared with the one generated with the double filter strategy (50%). On the other hand, the percentage of bundle sheath in the snRNA-seq library generated with the single filter strategy cells was lower (8.6%) compared with the one generated with the double filter strategy (15.2%). The remanding of the cell types showed small variations between both libraries, around 1%.

We also observed a significant difference in the number of cells for cluster 3 (which corresponds to mesophyll cells). The number of cells is significantly higher in the sample prepared with the single filter strategy (636 cells) compared with the sample generated with the double filter strategy (53 cells). Further analysis showed that the average number of UMIs and the number of genes per cell in cluster 3 also differ significantly. The snRNA-seq library prepared with the single filter strategy (higher chloroplast contamination) had an average number of UMIs and number of genes per cell of 2,029.53 and 1,261.9, respectively, whereas the snRNA-seq library prepared with the double filter strategy (lower chloroplast contamination) had an average number of UMIs and number of genes per cell of 5,572.83 and 2,492.19, respectively.

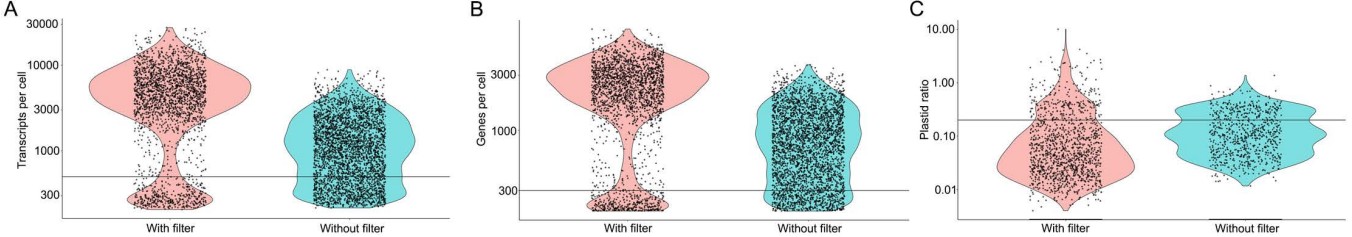

**Fig 4. Comparison of standard quality control metrics.** (A) Number of transcripts per cell identified for both libraries generated with and without the removal of chloroplast contamination. (B) Number of genes identified per cell for both libraries generated with and without the removal of chloroplast contamination. (C) Percentage of chloroplast-encoded genes per cell for both libraries generated with and without removal of chloroplast contamination.

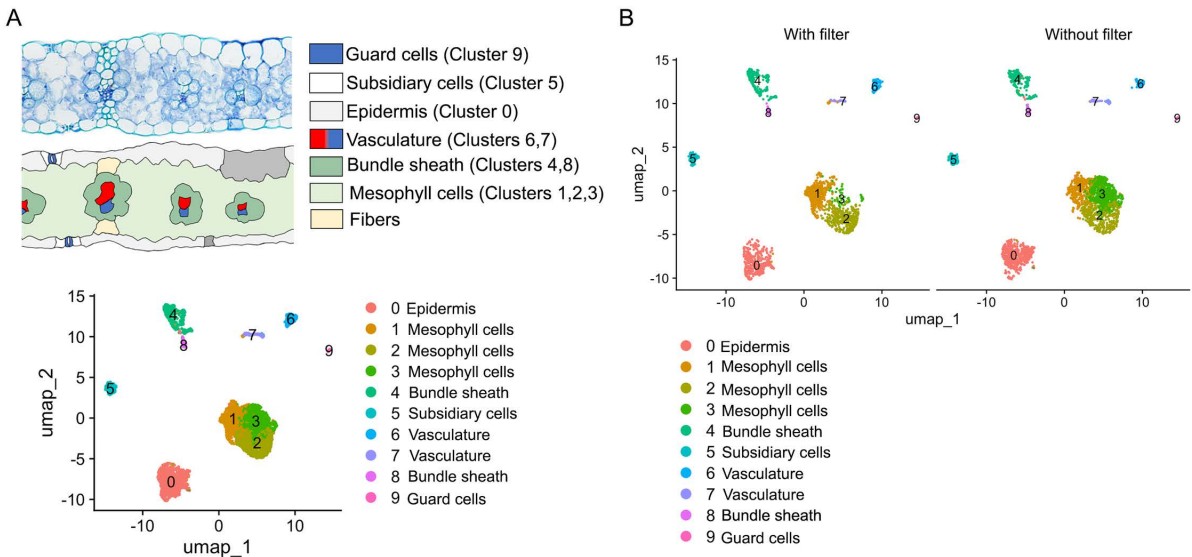

**Fig 5. Impact on cell clustering and cell type identification.** (A) Cross-section and graphical representation of the different cell types in the maize leaf. UMAP plot showing the clustering of the cells based on their gene expression profiles for (A) integrated data, (B) the library processed with our method of double filter strategy, and (C) the library processed with a previously published method of single filter strategy. Each dot corresponds to a cell, and they are colored based on their cluster identity.

**Table 2. Comparison of cell type composition.**

|  | Mesophyll | Bundle sheath | Subsidiary cells | Guard cells | Vasculature | Epidermis |
|---|---|---|---|---|---|---|
| Double FACS Filter | 50% | 15.2% | 5.3% | 0.6% | 6% | 22.9% |
| Single FACS Filter | 58.4% | 8.6% | 4.4% | 0.9% | 6.4% | 21.3% |

To confirm the effectiveness of our method in significantly decreasing chloroplast contamination during nuclei isolation, we isolated nuclei and performed FACS using both the single and double filter strategies separately. We counted the number of nuclei and chloroplasts in both samples (Fig 6A). As expected, the number of chloroplasts was significantly lower in the sample prepared with our protocol. In addition, we extracted the total RNA from the nuclei isolated using the single filter strategy (higher chloroplast contamination) and the double filter strategy (lower chloroplast contamination). To confirm lower chloroplast contamination in the sample prepared with our proposed method, we performed real-time quantitative PCR using primers for two chloroplast-encoded genes (*atpA* and *psbA*). The expression of both genes was significantly lower in the sample prepared with our proposed method (Fig 6B). Together, these results confirm that our proposed method is efficient in decreasing chloroplast contamination during nuclei isolation from tissues with a high presence of chloroplasts.

## Discussion

Protocols designed for nuclei isolation in the context of single-cell transcriptomics must carefully consider the unique characteristics of each tissue under investigation to ensure the extraction of high-quality nuclei and maintain overall data quality. Currently, there is a lack of tissue-specific protocols, with the majority being tested in root tissues [9,13,31,32]. These protocols are applicable across various tissues, but their universal nature overlooks tissue-specific differences that can influence the attainment of high-quality nuclei and, consequently, impact the quality of the resulting datasets.

…
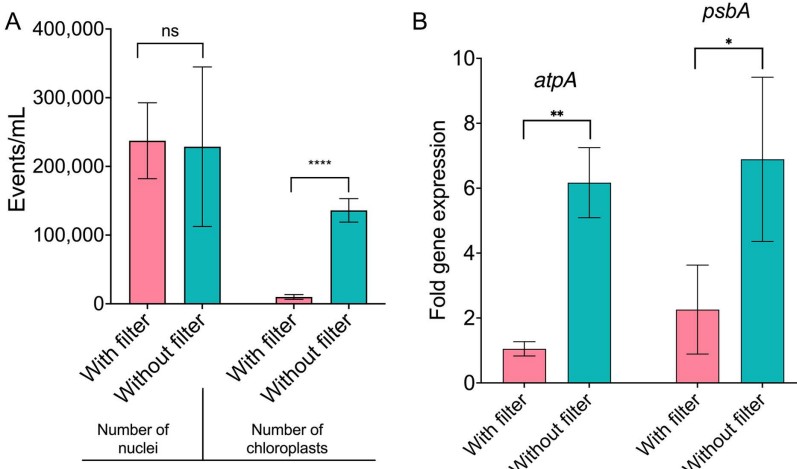

**Fig 6. Effectiveness of the proposed method to decrease chloroplast contamination.** (A) Counting of the number of nuclei and chloroplasts in samples generated with the double filter strategy (with filter) and single filter strategy (without filter). (B) Relative expression of *atpA* and *psbA*. Relative transcript levels were quantified by RT-qPCR and normalized with the housekeeping genes *EIF1α* and *Ubq7* using the $2^{-\Delta\Delta Ct}$ method. Error bars, SD. Mann-Whitney test was used to determine statistical significance; n = 3. Significance is indicated by asterisks: (*) *P*-value < 0.05; (**) *P*-value < 0.01; (****) *P*-value < 0.0001.

In the present work, we have developed an improved nuclei isolation protocol for leaf tissue and tested it in maize leaves. We incorporated a stringent FACS strategy that accounts for the high presence of chloroplasts in leaf tissue that can interfere with downstream analysis. The presence of plastids in the nuclei suspension can lead to data loss and impact factors such as sequencing quality and library complexity. The presence of chloroplast-encoded genes in the snRNA-seq dataset contributes to the pool of "low quality" potential nuclei, which, in turn, can affect sample size and compromise the statistical robustness of subsequent analyses.

With our protocol, the fraction of reads confidently mapped to the genome increased by almost 14%, an important increase compared to the sample prepared with a previously published protocol [8]. Furthermore, the reads mapped to the transcriptome had a similar increase using our improved double filter strategy. These improvements can directly impact the downstream analysis as increasing them allows for better detection of genes and cell type identification.

By applying the method presented here, we reduced the presence of chloroplast in the sample, although some cells still presented plastid transcripts. The portion of cells that have a very low percentage of these transcripts is much greater in the sample that was prepared with the double filter strategy compared to the sample prepared with the single filter strategy. In addition, the transcripts and genes per cell metrics were improved. All these improvements contributed to the fact that after the quality control thresholds were applied, the library created with the double filter strategy lost almost half the number of cells compared to the library prepared with the single filter strategy. These results show the potential of the protocol presented here to improve the quality of the datasets acquired from leaf tissues by effectively reducing the presence of chloroplast contamination and thus contributing to the improvement of sequencing quality. The acquisition of high-quality datasets is of vital importance for the success of snRNA-seq experiments. This holds particular significance when analyzing low-abundance cell types or exploring plant species that are less studied or present heightened challenges in nuclei isolation. In instances where cell numbers are inherently low, any incremental increase in dataset size contributes significantly to enhancing the power of discovery.

To verify that the more stringent protocol is suitable for its application in single-nuclei transcriptomics analysis, we applied the standard Seurat pipeline to both datasets after quality control. The resulting UMAP plots did not show any

negative impact attributable to our method, as we observed similar clustering patterns for both datasets and a good resolution of the cellular heterogeneity. In addition, we were able to confidently identify most of the cell types present in maize leaves using cell-type specific markers in both data sets. These results suggest that applying a more stringent nuclei isolation protocol may not impact downstream analysis for leaf tissues, and our protocol would be a viable alternative for the improvement of studies being done in tissues with high content of chloroplasts. Notably, we observed variations in the proportion of cells identified as mesophyll and bundle sheath. In contrast to the sample prepared with a single filter strategy (higher chloroplast contamination), our current method yielded a lower percentage of cells identified as mesophyll and an increased percentage of cells identified as bundle sheath. Both cell types represent the photosynthetically active cells of the leaf and are enriched in chloroplast content, and maize plants were shown to have the same number of chloroplasts in mesophyll and bundle sheath cells [33]. A potential explanation for the discrepancy in the number of mesophyll cells is the incomplete removal of cytoplasm from the nuclei of these cells, leaving more chloroplasts attached to the nuclei. Another explanation could be that chloroplasts, instead of nuclei, were partitioned into droplets with the Gel Bead-In Emulsions (GEMs) during the barcoding step of the 10X Single-cell library preparation. These chloroplasts might be subsequently removed during the initial step of our sorting strategy. However, it is important to acknowledge that the samples were not grown and collected simultaneously, although we controlled for growth conditions, developmental stage, and time of day. Therefore, there could be inherent biological differences between the samples, and the observed variations may not necessarily stem from a bias in our method but rather expected differences between individual plants. This could explain the increase in bundle sheath since if the cause were the removal of cells due to the close association of nuclei with chloroplasts, one would anticipate a similar impact on both mesophyll and bundle sheath cells. Nonetheless, it's noteworthy that the overall proportions of cell types remain consistent between both samples, with mesophyll cells being the most abundant and guard cells the least present, aligning with our expectations.

Overall, our protocol is an alternative approach to generating good-quality snRNA-seq datasets for leaf tissues. The ability to remove chloroplasts without affecting nuclear integrity allowed us to improve the overall quality of our dataset. We believe this protocol could be of great value for obtaining good-quality datasets from chloroplast-rich tissues in non-model or hard-to-work-with species, where data acquisition is generally harder and there is less room for loss of quality. In addition, this protocol could be incorporated into other single-nuclei technologies like single-cell ATAC-seq.

## Supporting information

**S1 Table. Cell-type-specific markers for cell type identification of maize leaf snRNA-seq.**
(PDF)

**S2 Table. Primer sequences.**
(PDF)

**S3 Table. Number of cells per cluster for each sample.**
(PDF)

**S1 Fig. Summary of the distribution of the number of genes per nucleus.** (A) Histogram of the number of genes per nucleus in the sample prepared with the FACS single-filter strategy (higher chloroplast contamination). (B) Histogram of the number of genes per nucleus in the sample prepared with the FACS double-filter strategy (lower chloroplast contamination). (C) Summary of the quantile summary of the number of genes per nucleus.
(TIFF)

**S2 Fig. Dot plot of cell-type-specific marker expression per cluster.**
(TIFF)

**S3 Fig. Effectiveness of FACS double filtering strategy in removing chloroplast contamination.** (A) Dot-plot of PerCP and DAPI positive events. The black dots indicate PerCP-positive particles. (B) Dot-plot of PerCP and DAPI positive events. Orange dots correspond to DAPI-positive events.
(TIFF)

## Acknowledgments

We thank Christopher Dervinis for his advice and insightful discussions throughout the course of this study. We thank Alexander Linares for the technical advice and assistance in the development of the FACS protocol in the ICBR Cytometry Core (RRID:SCR_019119). We also thank Dr. Robert Ferl and Dr. Ana-Lisa Paul from the Space Biology laboratory at the University of Florida for kindly letting us use the microscope.

## Author contributions

**Conceptualization:** Kelly Mayrink Balmant, Gabriela Madrid, Marcio F.R. Resende Jr..

**Formal analysis:** Gabriela Madrid, Gabriel Angelo Saraiva Raimundo, Fabian Andres Reyes.

**Funding acquisition:** Marcio F.R. Resende Jr..

**Investigation:** Gabriel Angelo Saraiva Raimundo, Fabian Andres Reyes.

**Methodology:** Gabriela Madrid, Fabian Andres Reyes.

**Resources:** Kelly Mayrink Balmant.

**Supervision:** Kelly Mayrink Balmant, Marcio F.R. Resende Jr..

**Validation:** Gabriel Angelo Saraiva Raimundo, Fabian Andres Reyes, Edgard Augusto de Toledo Picoli.

**Writing – original draft:** Kelly Mayrink Balmant, Gabriela Madrid, Marcio F.R. Resende Jr..

**Writing – review & editing:** Kelly Mayrink Balmant, Gabriela Madrid, Marcio F.R. Resende Jr., Gabriel Angelo Saraiva Raimundo, Fabian Andres Reyes, Edgard Augusto de Toledo Picoli.

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
