## [Decision Letter · Decision Letter 0]

9 May 2024

Dear Dr. Balmant,

Thank you for submitting your manuscript to PLOS ONE. After careful consideration, we feel that it has merit but does not fully meet PLOS ONE’s publication criteria as it currently stands. Therefore, we invite you to submit a revised version of the manuscript that addresses the points raised during the review process.

We look forward to receiving your revised manuscript.

Kind regards,

Tzen-Yuh Chiang

Academic Editor

PLOS ONE

Journal Requirements:

"We thank Christopher Dervinis for his advice and insightful discussions throughout the course of this study. Funding was provided by USDA NIFA SCRI (Grant No. 2018-51181-28419 and 2022-51181-38333)."

"MFRR

USDA NIFA SCRI Grant No. 2018-51181-28419 and 2022-51181-38333

United States Department of Agriculture National Institute of Food and Agriculture Specialty Crop Research Initiative

https://www.nifa.usda.gov/grants/funding-opportunities/specialty-crop-research-initiative

The the sponsors or funders did not play any role in the study design, data collection and analysis, decision to publish, or preparation of the manuscript"

Reviewers' comments:

Reviewer's Responses to Questions

**Comments to the Author**

1. Is the manuscript technically sound, and do the data support the conclusions?

Reviewer #1: No

2. Has the statistical analysis been performed appropriately and rigorously?

Reviewer #1: No

3. Have the authors made all data underlying the findings in their manuscript fully available?

Reviewer #1: No

4. Is the manuscript presented in an intelligible fashion and written in standard English?

Reviewer #1: No

Reviewer #1: Please see the attached PDF for my full review comments.

Also, I did not find the link to access any of the data; it is not found in the github repository either. Please check whether you have included it in the manuscript.

**Do you want your identity to be public for this peer review?** For information about this choice, including consent withdrawal, please see our Privacy Policy

Reviewer #1: No

---

## [Author Response · Author response to Decision Letter 1]

11 Dec 2024

All the reviewers comments were addressed in the file Response to reviewers.

---

## [Decision Letter · Decision Letter 1]

18 Feb 2025

Dear Dr. Balmant,

Thank you for submitting your manuscript to PLOS ONE. After careful consideration, we feel that it has merit but does not fully meet PLOS ONE’s publication criteria as it currently stands. Therefore, we invite you to submit a revised version of the manuscript that addresses the points raised during the review process.

We look forward to receiving your revised manuscript.

Kind regards,

Tzen-Yuh Chiang

Academic Editor

PLOS ONE

Journal Requirements:

Reviewers' comments:

Reviewer's Responses to Questions

**Comments to the Author**

Reviewer #2: All comments have been addressed

Reviewer #3: All comments have been addressed

2. Is the manuscript technically sound, and do the data support the conclusions?

Reviewer #2: Partly

Reviewer #3: Partly

3. Has the statistical analysis been performed appropriately and rigorously?

Reviewer #2: No

Reviewer #3: Yes

4. Have the authors made all data underlying the findings in their manuscript fully available?

Reviewer #2: No

Reviewer #3: Yes

5. Is the manuscript presented in an intelligible fashion and written in standard English?

Reviewer #2: Yes

Reviewer #3: Yes

Reviewer #2: In a revised manuscript to PLoS One, Madrid and colleagues present their findings regarding a DAPI+/Chlorophyll- sorting strategy for nuclei prior to 10x genomics microfluidic library preparation. Using this strategy the authors convincingly show depletion of autofluorescence (presumed to be chlorophyll), depletion of cpDNA, and marked atpA and psbA demonstrated independently by qPCR. In response to the first round of reviews, of which I read but was not involved, the authors appear to have added descriptive statistics to their results and several language changes suggested by previous reviewers. In my reading of this, my opinion is that the findings are meaningful but could benefit from more precise language especially pertaining to how the presence of chlorophyll and/or ambient RNA could impact the results when present.

Major concerns/comments:

1. The strongest data in this manuscript appears to be the removal of cpDNA and the qPCR data regarding atpA and psbA. I find this sufficient as is, but disconnected from mentions of statistical power, identification of cell type identity, or any other snRNAseq applications. Are there specific applications for performing snRNAseq that are disrupted by chlorophyll presence (diff expression, a cell type that is depleted, any other analysis)? Without demonstrating that the DAPI+/autoflu- fixes that, the only improvement that I understand is increased reads mapping to genome (which I would note is low even in the improved condition).

2. The observations appear to be made from running 2 samples (one from cell suspension without FACS and one with FACS “double” meaning “DAPI+/autoflu-“ is that the case? The authors do a fine job reporting QC statistics, but including sample size and replicates are necessary to making claims that this is a “novel sorting strategy” that others should employ.

3. I am unable to assess the rigor of data analysis as the data nor the scripts were made accessible to me. Github link points to an account, but the accompanying code for this manuscript was not obvious to find. Please update along with GEO or other database references.

Minor concerns/comments:

• In “computational workflow for scRNAseq data analysis” on line 197, how many barcodes/nuclei were removed with the <0.2% plastid percentage? Were less removed from the FACS filtered? This would be critical evidence to the authors points. Same question for the 0.5% chloroplast perentage in line 214 under “Generation of quality metrics”.

• Descriptions of ambient RNA were made however, there are several existing packages that could predict presence of ambient RNA and report statistics to help the author’s points. It would be of interest if these algorithms predict higher ambient RNA in non-FACS sample(s) compared to FACS.

Reviewer #3: Gabriela Madrid and colleagues developed a very efficient protocol to remove chloroplasts from leaf tissue of plant during the snRNA-seq experiment. They set up a new method of applying double filters strategy which first negatively select autofluorescent chloroplast and then positively select DAPI stained nuclei, thus achieving a relatively purified nuclei and snRNA-seq data.

While I appreciate the value of this research, I think there are some aspects of this manuscript need to be further polished and more analysis should be done to deliver a more solid conclusion:

1. Line 47, it is a little bit confusing to me to say "The success of scRNA-seq is dependent on the acquisition of high-quality datasets." As a researcher most working on wet-lab experiment, the aim of scRNA-seq experiment, at least to myself, is to obtain high quality data. It is a little bit weird to say "the success of scRNA-seq depends on acquiring high quality datasets." Sounds like attributing result as reason.

2. Line 82-84, "For droplet-based methods, where the individual cells or nuclei are captured in microfluidic droplets containing the reagents for library preparation, it is assumed that each droplet contains only RNA from an intact single cell." It would be more precise to add "or nucleus" at the end of this sentence.

3. Line 152, it should be "double filters" or "double-filter" strategy.

4. Figure 3A, previous introduction mentioned that DAPI could also stain chloroplasts causing contaminations during FACS sorting. But in this experiment, there is no obvious clue that chloroplasts are also stained by DAPI. Need more discussion to explain the reason of the difference between FACS and microscope experiment.

5. Line 373-377, the qPCR result of atpA and psbA from extracting RNA with different purification strategies, I think it would be better to put this experiment and data in the first two figures, that demonstrates the effectiveness of double filters strategy with a relatively simple but less comprehensive way.

**Do you want your identity to be public for this peer review?** For information about this choice, including consent withdrawal, please see our Privacy Policy

Reviewer #2: **Yes: ** Dominic J. Acri

Reviewer #3: No

---

## [Author Response · Author response to Decision Letter 2]

3 Jun 2025

Reviewer #2:

In a revised manuscript to PLoS One, Madrid and colleagues present their findings regarding a DAPI+/Chlorophyll- sorting strategy for nuclei prior to 10x genomics microfluidic library preparation. Using this strategy the authors convincingly show depletion of autofluorescence (presumed to be chlorophyll), depletion of cpDNA, and marked atpA and psbA demonstrated independently by qPCR. In response to the first round of reviews, of which I read but was not involved, the authors appear to have added descriptive statistics to their results and several language changes suggested by previous reviewers. In my reading of this, my opinion is that the findings are meaningful but could benefit from more precise language especially pertaining to how the presence of chlorophyll and/or ambient RNA could impact the results when present.

Major concerns/comments:

1. The strongest data in this manuscript appears to be the removal of cpDNA and the qPCR data regarding atpA and psbA. I find this sufficient as is, but disconnected from mentions of statistical power, identification of cell type identity, or any other snRNAseq applications. Are there specific applications for performing snRNAseq that are disrupted by chlorophyll presence (diff expression, a cell type that is depleted, any other analysis)? Without demonstrating that the DAPI+/autoflu- fixes that, the only improvement that I understand is increased reads mapping to genome (which I would note is low even in the improved condition).

Maize mesophyll cells, which are the most abundant cell type in maize leaves, contain an average of 53 chloroplasts per cell. In other words, a mesophyll cell has a chloroplast-to-nucleus ratio of 53:1 (Lee et al., 2023). As mentioned in the manuscript (Lines 27-28), DAPI can also bind to the plastid genome, causing these organelles to be classified as nuclei due to a positive DAPI signal. Since leaf cells contain a significantly higher number of chloroplasts per cell, in the positive events during FACS, there will be more chloroplasts than nuclei. This is corroborated by the results described on lines 269-272: “Out of 2.5M initial events, approximately 1.4M events were gated as a PerCP-negative population to continue with the FACS strategy (Fig 1A), and from this population of events, DAPI-stained particles were selected for further analysis (~15,000 events) (Fig 2B, C).” This indicates that 44% of the initial events were eliminated due to autofluorescence signals and potential contamination from chloroplasts.

By using the double-filter strategy, we were able to reduce the contamination of chloroplasts. This can be confirmed not only by the improvement in the alignment rate (from 54% to 68%), but also in the number of genes detected. In the sample prepared with the single-filter strategy (higher chloroplast contamination), we detected 12,607 genes. On the other hand, we detected 16,886 genes in the sample prepared with the double-filter strategy (lower chloroplast contamination). This information has been added to the revised manuscript (lines 352-355).

In conclusion, our double-filter strategy for reducing chloroplast contamination during nuclei isolation for snRNA-seq experiments not only improves the alignment rate but also increases the number of detected genes. It is important to note that by increasing the number of detected genes, we may also enhance the chances of identifying low-expressed genes.

Lee M, Boyd RA, Boateng KA, Ort DR (2023) Exploring 3D leaf anatomical traits for C4 photosynthesis: chloroplast and plasmodesmata pit field size in maize and sugarcane. New Phytologist 239(2):506-517.

2. The observations appear to be made from running 2 samples (one from cell suspension without FACS and one with FACS “double” meaning “DAPI+/autoflu-“ is that the case? The authors do a fine job reporting QC statistics, but including sample size and replicates are necessary to making claims that this is a “novel sorting strategy” that others should employ.

Single-cell experiments are very costly (around $2,500 per sample). Although we lacked replicates for the single-cell assays, we were able to assess the effectiveness of our method (reducing chloroplast contamination) by repeating the FACS step three times and summarizing the statistics of events captured for both techniques (Fig. 6A – counting the number of chloroplasts and nuclei in samples obtained with the double-filter strategy [new method] and the single-filter strategy [old method]). We also conducted an RT-qPCR on the samples obtained using both methods to examine the expression of chloroplast-encoded genes (Fig. 6B).

We believe the chloroplast counting from three different independent replicates provides strong evidence that the difference we observe between the two methods (double-filter versus single-filter strategy) is not due to batch effect. We agree that more replicates of snRNA-seq would provide additional evidence, but we believe this should be done in future studies, considering the cost involved and the fact that the different sources of evidence provided here support the proposed method.

3. I am unable to assess the rigor of data analysis as the data nor the scripts were made accessible to me. Github link points to an account, but the accompanying code for this manuscript was not obvious to find. Please update along with GEO or other database references.

- All raw and processed data have been submitted to the NCBI Gene Expression Omnibus under accession number GSE297213.

To review GEO accession GSE297213:

Go to https://nam10.safelinks.protection.outlook.com/?url=https%3A%2F%2Fwww.ncbi.nlm.nih.gov%2Fgeo%2Fquery%2Facc.cgi%3Facc%3DGSE297213&data=05%7C02%7Cbalmant%40ufl.edu%7C301b5e9e4da2496d006808dda1e8ddb7%7C0d4da0f84a314d76ace60a62331e1b84%7C0%7C0%7C638844743702227738%7CUnknown%7CTWFpbGZsb3d8eyJFbXB0eU1hcGkiOnRydWUsIlYiOiIwLjAuMDAwMCIsIlAiOiJXaW4zMiIsIkFOIjoiTWFpbCIsIldUIjoyfQ%3D%3D%7C0%7C%7C%7C&sdata=ctvoX2hSVLB8Q%2BLWjFRGdAb8X9L4whM549sUmATK5v0%3D&reserved=0

Enter token kzinoeoilzallcn into the box

- Custom scripts are available at https://github.com/Resende-Lab in a folder called Madrid_etal_2025.

Minor concerns/comments:

• In “computational workflow for scRNAseq data analysis” on line 197, how many barcodes/nuclei were removed with the <0.2% plastid percentage? Were less removed from the FACS filtered? This would be critical evidence to the authors points. Same question for the 0.5% chloroplast percentage in line 214 under “Generation of quality metrics”.

We apologize for the confusion. The parameter used for chloroplasts was that potential nuclei with a chloroplast percentage greater than 0.5% were excluded from further analysis. The value of 0.2% on line 500 was a typo, and we have corrected it in the updated version of the manuscript.

After applying the threshold, 479 barcodes (corresponding to 19% of the total number of barcodes) were removed from the dataset generated using the double filter strategy, while 844 barcodes (corresponding to 33% of the total number of barcodes) were removed from the dataset prepared with the single filter strategy. This information was included in the revised manuscript.

• Descriptions of ambient RNA were made however, there are several existing packages that could predict presence of ambient RNA and report statistics to help the author’s points. It would be of interest if these algorithms predict higher ambient RNA in non-FACS sample(s) compared to FACS.

We used SoupX to determine the level of contamination in both samples and found that the sample prepared with the FACS double-filter strategy (our method) has a contamination fraction of 8%. The sample prepared with the FACS single-filter method has a contamination fraction of 12%. We used three marker genes for the bundle sheath cell type to calculate the contamination fraction (Zm00001eb121470, Zm00001eb092540, and Zm00001eb197410).

Reviewer #3:

Gabriela Madrid and colleagues developed a very efficient protocol to remove chloroplasts from leaf tissue of plant during the snRNA-seq experiment. They set up a new method of applying double filters strategy which first negatively select autofluorescent chloroplast and then positively select DAPI stained nuclei, thus achieving a relatively purified nuclei and snRNA-seq data.

While I appreciate the value of this research, I think there are some aspects of this manuscript need to be further polished and more analysis should be done to deliver a more solid conclusion:

1. Line 47, it is a little bit confusing to me to say "The success of scRNA-seq is dependent on the acquisition of high-quality datasets." As a researcher most working on wet-lab experiment, the aim of scRNA-seq experiment, at least to myself, is to obtain high quality data. It is a little bit weird to say "the success of scRNA-seq depends on acquiring high quality datasets." Sounds like attributing result as reason.

Thanks for the comment. We have changed the sentences to: “The success of scRNA-seq is dependent on the acquisition of high-quality cells."

2. Line 82-84, "For droplet-based methods, where the individual cells or nuclei are captured in microfluidic droplets containing the reagents for library preparation, it is assumed that each droplet contains only RNA from an intact single cell." It would be more precise to add "or nucleus" at the end of this sentence.

Thanks for the suggestion. Change has been made in the revised manuscript.

3. Line 152, it should be "double filters" or "double-filter" strategy.

Thanks for the suggestion. Change has been made in the revised manuscript.

4. Figure 3A, previous introduction mentioned that DAPI could also stain chloroplasts causing contaminations during FACS sorting. But in this experiment, there is no obvious clue that chloroplasts are also stained by DAPI. Need more discussion to explain the reason of the difference between FACS and microscope experiment.

As stated in the manuscript, DAPI can bind to the plastid genome (Jemes & Jope, 1979; Rowan et al., 2007). signal

The main reason why Fig. 3A does not show chloroplasts with a DAPI signal because of the sensitivity of the confocal. FACS generally exhibits higher sensitivity than confocal microscopes for DAPI signal detection. IN the case of the confocal microscope, the sensitivity is often limited by the background fluorescence.

Another factor to consider is how the data is presented and what defines a signal. Generally, FACS is the more sensitive assay, with data displayed on a log scale. In contrast, imaging (confocal microscopy) is somewhat binomial—you determine what is positive based on visual assessment. You will adjust the laser power and exposure time accordingly before acquiring the image. Even when images are obtained in 16-bit format, most of the extended dynamic range appears black, and the signals that are visible are predominantly linear. This means that signals that seem to be mid-range by flow can appear black in imaging (confocal or fluorescent microscopy.

In addition, the maize nuclear genome (2.4Gb) is ~17,000 larger than the chloroplast genome (140,384 bp). This could also affect this difference.

Please see below, images where we increased the exposition time and were able to see some DAPI-positive signal in the chloroplasts.

Maize – Analyzed on the fluorescence microscope. (A) GFP filter, (B) DAPI filter. Arrows indicate chloroplasts.

Bienertia sinuspersici – Analyzed on the confocal microscope (A) Bright field, (B) DAPI filter, (C) GFP filter. Arrows indicate chloroplasts.

James TW, Jope C (1979) Visualization by fluorescence of chloroplast DNA in higher plants by means of the DNA-specific probe 4'6-diamidino-2-phenylindole. J Cell Biol 79(3):623-630.

Rowan, B.A., Oldenburg, D.J. & Bendich, A.J. (2007) A high-throughput method for detection of DNA in chloroplasts using flow cytometry. Plant Methods 3, 5

5. Line 373-377, the qPCR result of atpA and psbA from extracting RNA with different purification strategies, I think it would be better to put this experiment and data in the first two figures, that demonstrates the effectiveness of double filters strategy with a relatively simple but less comprehensive way.

Thanks for the suggestion, but we respectfully disagree with the reviewer and think that it makes more sense to have the data as Figure 6.

---

## [Decision Letter · Decision Letter 2]

27 Jun 2025

An improved nuclei isolation protocol from leaf tissue for single-cell transcriptomics

PONE-D-24-12593R2

Dear Dr. Balmant,

We’re pleased to inform you that your manuscript has been judged scientifically suitable for publication and will be formally accepted for publication once it meets all outstanding technical requirements.

Kind regards,

Tzen-Yuh Chiang

Academic Editor

PLOS ONE

Additional Editor Comments (optional):

Reviewers' comments:

Reviewer's Responses to Questions

**Comments to the Author**

Reviewer #2: All comments have been addressed

2. Is the manuscript technically sound, and do the data support the conclusions?

Reviewer #2: Yes

3. Has the statistical analysis been performed appropriately and rigorously?

Reviewer #2: Yes

4. Have the authors made all data underlying the findings in their manuscript fully available?

Reviewer #2: Yes

5. Is the manuscript presented in an intelligible fashion and written in standard English?

Reviewer #2: Yes

Reviewer #2: Upon reviewing the responses to reviewers, it is my opinion that the authors provided not only sufficient but extremely convincing answers to all concerns (from both reviewers).

The GitHub repo "Madrid_etal_2025" listed in the response to reviewers was not made public so I am unable to review what I asked for, however the authors responses and intent to make it public should be sufficient for publication in this journal.

Lastly, cost is not a sufficient justification for sample size. I recognize it is a limitation so do not feel strongly that the authors need to change any their current manuscript. We can agree to disagree on the use of statistics for single cell as the field has not reached a consensus. Furthermore, the FACS experiments are appropriately powered and convincing that the tech introduced by this manuscript represents a meaningful contribution to the field at large.

**Do you want your identity to be public for this peer review?** For information about this choice, including consent withdrawal, please see our Privacy Policy

Reviewer #2: No

---

## [Editor Report · Acceptance letter]

PONE-D-24-12593R2

PLOS ONE

Dear Dr. Balmant,

I'm pleased to inform you that your manuscript has been deemed suitable for publication in PLOS ONE. Congratulations! Your manuscript is now being handed over to our production team.

Kind regards,

on behalf of

Dr. Tzen-Yuh Chiang

Academic Editor

PLOS ONE